# Nature of Luminescence and Pharmacological Activity of Sulfaguanidine

**DOI:** 10.3390/molecules28104159

**Published:** 2023-05-18

**Authors:** Olga Tchaikovskaya, Elena Bocharnikova, Olga Bazyl, Vlada Chaidonova, George Mayer, Paul Avramov

**Affiliations:** 1Quantum Electronics Laboratory, Institute of Electrophysics, Ural Branch of the Russian Academy of Sciences, 620146 Yekaterinburg, Russia; 2Laboratory of Photophysics and Photochemistry of Molecules, Faculty of Physics, Tomsk State University, 634050 Tomsk, Russia; bocharnikova.2010@mail.ru (E.B.); olga.k.bazyl@gmail.com (O.B.); krayvlada0523@mail.ru (V.C.); mayer_gv@mail.tsu.ru (G.M.); 3Hygienic and Epidemiological Center in Republic of Khakassia, 655017 Abakan, Russia; 4Department of Chemistry and Green-Nano Materials Research Center, Kyungpook National University, 80 Daehak-ro, Buk-gu, Daegu 41566, Republic of Korea

**Keywords:** sulfaguanidine, spectroscopy, H-bonded complexes, computational chemistry

## Abstract

Sulfonamides are one of the oldest groups of veterinary chemotherapeutic agents. Physico-chemical properties, the concentration and the nature of the environment are the factors responsible for the distribution of sulfonamides in the living organism. Although these drug compounds have been in use for more than half a century, knowledge about their behavior is still limited. Physiological activity is currently attributed to the sulfanyl radical. Our study is devoted to the spectral properties of aqueous solutions of sulfaguanidine, in which the formation of complexes with an H-bond and a protonated form takes place. The nature of the fluorescent state of sulfaguanidine was interpreted using computational chemistry, the electronic absorption method and the luminescence method. The structure of sulfaguanidine includes several active fragments: aniline, sulfonic and guanidine. To reveal the role of fragments in the physiological activity of the studied antibiotic, we calculated and compared the effective charges of the fragments of aniline and sulfaguanidine molecules. Chromophore groups were identified in molecules, which determine the intermolecular interaction between a molecule and a proton-donor solvent. The study also revealed the impact of sulfone and guanidine groups, as well as complexation, on the effective charge of the antibiotic fragment responsible for physiological activity and luminescent ability.

## 1. Introduction

The modeling and prediction of the properties and biological activity of molecules depends on the choice of molecular representation [1]. Decades of research have been required to establish structure–activity and structure–property relationships for organic compounds. Understanding these relationships opens the way for improving the predictive modeling of biological activity and the properties of small molecules for the synthesis of new drugs. The monitoring of therapeutics in biological systems and foodstuffs holds great potential for improving patient outcomes and dramatically reducing healthcare costs [2,3,4]. Although it has been on the radar of the scientific community for nearly two decades, sensor-based approaches have not yet been widely adopted into clinical laboratory diagnostics, perhaps due to a gap between the scientific, industrial, and medical communities. Looking ahead, the molecular understanding of the spectral properties of molecules will become increasingly important and will provide a solid basis for future predictive tasks associated with the determination of drugs in serum and foods.

Modern computer technologies, bioinformatics and new experimental methods in the field of medicinal chemistry have provided the acceleration and optimization of the process of finding new biologically active compounds [1]. Currently, the pharmaceutical industry widely uses molecular modeling methods to study the relationship between the structure and the activity of molecules. Existing quantum chemical methods help to create new drugs by identifying the substituents and their locations in the molecule that are necessary to obtain a pharmaceutical effect. Among amine derivatives, there are a number of well-known highly effective drugs [5]. Very often, amines are not used in a free form but in the form of sulfonyl derivative or sulfanilamide preparations used to treat infectious diseases, which are mainly of bacterial origin. The identification of the most probable conformations for binding the molecule to the environment includes two stages: the study of the entire conformational space of the molecular structure and the exact determination of the site of interaction. To search for the corresponding conformations of the molecule, systematic and stochastic search methods are used. These methods give variations in structural parameters (angles and bond lengths), gradually revealing the appropriate conformation [6].

With the current variety of antimicrobial drugs, sulfonamides [7,8,9], which are used in medicine [10,11,12,13] and veterinary medicine, still play an important role in the fight against and prevention of various types of infections, having antibacterial, hypoglycemic and other types of biological activity. Despite the diverse number of new antibacterial agents, sulfonamides are regularly prescribed for the treatment of various infectious diseases, type 2 diabetes mellitus [7,14,15,16,17] and urinary tract infections [18] and for first aid, using napkins from hemostatic collagen plates in combination with zeolite powder and antibacterial drugs [19]. Typical representatives of sulfonamides are sulfanilamide, sulfacyl sodium, urosulfan, sulgin, norsulfazol, phthalazol, etazol, sulfadimezin, sulfapyridazine, sulfamonomethoxine, sulfadimethoxine, sulfalene and saladosin.

N-(4-aminobenzenesulfonic) guanidine or sulfaguanidine (Figure 1) is the active ingredient of the medicine Sulgin, a derivative of guanidine sulfanilamide, which was independently obtained by Marshall, Bratton, White, Lichfield and Roblin, Williams, Winneck and English in 1940 [20]. It was first introduced for the treatment of bacillary dysentery. This treatment is facilitated by its poor adsorption in the proximal colon, which helps it reach the distal colon, where the main disease occurs [21]. Its oral route of administration is now well established [22].

Due to the high antimicrobial and antibacterial activity, sulfaguanidine is used as a part of various structures [23] to improve the mechanical and anticorrosion properties and increase the thermal stability of composites. Structural properties play an important role in the development of new alternative drugs based on sulfaguanidine [24,25]. Because of their usefulness in predicting cytotoxicity, hERG inhibition, hepatotoxicity and skin sensitization [26], spectrochemical and pharmacological properties are supported by existing guidelines for the design of future drugs [24]. For example, the larger the chain in the descriptor structure, the stronger the hERG inhibition [27]. In one study, the authors showed that the least inhibitory property has a drug containing >C=O or -O- groups in the structure. The amide/imide tautomerism of the molecules of sulfate preparations in their neutral form is widely discussed in the literature [28,29]. Studies have shown that in sulfonyl guanidine, the sulfonamide group is the preferred protonation site in a solution.

An important parameter of the drug is the bioavailability index, which is directly determined by the number of sites in the donor and acceptor molecules of hydrogen bonds. The structures of sulfaguanidine and its azaione were studied using both infrared spectra and DFT calculations [24,30]. The authors described strong spectral changes caused by the conversion of the sulfaguanidine molecule to the corresponding azanion. Structural changes, which are accompanied by manifestations in the spectra, are caused by the redistribution of the charge on the fragments and nearby in the environment. According to a data analysis, the maximum negative charge is concentrated on the sulfonyl guanidine fragment [31].

The purpose of the study is to study the active fragments of sulfaguanidine with specific properties, the effect of complexation and protonation on these properties by spectroscopy, and quantum chemical methods.

## 2. Results and Discussion

### 2.1. Conformational Design

A theoretical study of molecular systems in solutions was carried out due to the lack of reliable information about the spatial structure of the molecular system. In particular, the exact structure of the first coordination sphere of a molecule in a proton-donor solvent is unknown. We believe that the closer the chosen geometry of the studied molecule and the location of the solvent molecules in the first coordination sphere of the real molecule, the better the agreement between the calculated spectral-luminescent properties and the experimental data [32].

The antimicrobial activity of an antibiotic largely depends on two factors: the conformation (geometry) of its molecule and its chemical reactivity. The possibility of inscribing a physiologically active fragment of the antibiotic molecule into the active center of the target bacterium (the “key-lock” principle) depends on the first factor. Since any interaction between molecules, including chemical reactions, is determined at the first stage by the electrostatic interaction of these molecules, the assessment of chemical reactivity can initially be based on the calculated effective charges of fragments of the antibiotic molecule. Knowledge of the influence of various factors on the distribution of effective charges on antibiotic fragments is important because any chemical reaction is caused by the interaction of the electric charges of the reacting objects. The interaction of the charges of the fragments within the molecule itself affects its spatial structure, which provides the possibility of approaching the interacting objects at the required distance.

In our study, the criterion for the correct choice of the conformation of the molecule closest to the real one is the correspondence between the calculated absorption spectra and the experimental ones, which are in a nonpolar solvent for an isolated molecule and in a polar solvent for complexes. In our view, it is difficult to give priority to any of the above factors, the coincidence of which ensures the emergence of the antibacterial activity of the antibiotic. At the first stages of studying the physiological activity of the sulfonamide series, it was shown that the activity of the antibiotics of this series is determined by the aniline fragment of the molecule: the only common structure responsible for maintaining activity is the aniline fragment [33]. Studies [34] have shown that the activity of sulfa drugs is primarily determined by the properties of the amino group of the aniline fragment at position 4 and indirectly by the nature of the substituents in position 1.

When calculating the geometry of the ground state, X-ray diffraction data (lengths of chemical bonds and bond angles) are usually used, in addition to modern methods for optimizing the geometry of molecules [35,36].

The structure of sulfaguanidine (Figure 1) includes several active fragments: aniline, sulfonic and guanidine. To compare the changes in the activity of fragments in sulfaguanidine, we first obtained calculations of the electronic spectra of isolated aniline and its H-bonded and protonated complexes (Figure 2).

Table 1 lists the energies of the three lower electronically excited states of aniline S_1_, S_2_ and S_3(4)_. The experimental data for the aqueous solutions of aniline, according to [37,38,39], give two absorption bands with maxima at 280 and 230 nm. The first band, according to the calculation, is formed by the S_0_→S_1_ (ππ *) transition, and the second is formed by the S_0_→S_2_ (ππ *) transition. An intramolecular transfer of electron density accompanies electronic transitions from the amino group to the benzene ring. It is known that the amino group is characterized by a small pyramidally since the hybridization of the nitrogen atom lies between the sp^3^ and sp^2^ types. In our calculation for aniline, the height of the pyramid is assumed to be 0.25 Å. The exit of the nitrogen atom from the plane of molecules, as well as the formation of an H-bonded complex and a protonated form, reduces the symmetry of the molecule and leads to the appearance of an electronic transition, S_0_→S_4_, which, according to the calculation, lies in the region of 211–214 nm. Note that taking into account the pyramidal nature of the nitrogen atom of the amino group also leads to a shift of electronic transitions towards higher energies and a decrease in their intensity.

The formation of the protonated form of aniline (Figure 2b) with both planar and non-planar structures leads to a shift of the band in the middle region of the absorption spectrum from 250 nm for isolated aniline to the short-wavelength side of 230 nm (Table 2). This absorption band agrees with the experimental result for aniline in water.

The values of the minima of the electrostatic potential (Figure 3a) are −143 kJ/mol (coordinates: x = 4.2, y = 0.0, z = 1.1, Å) for planar aniline in the S_0_ state and −200 kJ/mol (coordinates: x = 4.2, y = 0.0, z = 1.6, Å) for non-planar aniline (Figure 2). This indicates that the place of interaction of aniline during the formation of the H-bonded complex and protonation is the nitrogen atom of the amino group (Table 2).

The calculations of an aniline molecule with planar and pyramidal amino groups showed that, in an isolated molecule in the ground state, the amino group exhibits electron-withdrawing properties, while the benzene ring exhibits donor properties. In general, the aniline molecule is neutral. The formation of a complex with an H-bond leads to an increase in electron density transfer, enhancing the acceptor properties of the amino group. The water molecule participates in weaknesses in the process of electron density transfer, exhibiting weaker electron acceptor properties than the amino group. In the protonated form, the acceptor of the amino group is enhanced even more with a weak participation in the transfer of the solvated proton (Table 2).

The differentiation of the contour of the absorption band of the experimental absorption spectrum of sulfaguanidine in water identifies the following maxima: 290, 259, 208, and 197 nm [40]. The non-planar structure of the sulfaguanidine molecule as a whole makes it difficult to determine the orbital nature of electronically excited states due to the mixing of σ- and π-type atomic orbitals. The absorption spectrum of sulfaguanidine, according to the calculations, is formed by the S_0_→S_1_ (ππ *) transition and two one-electron transitions of differing intensity and orbital nature S_0_→S_2_ (πσ *) and S_0_→S_3_ (ππ *) (Table 3). The intensity of the absorption band in the region of 259 nm forms mainly the S_0_→S_3_ (ππ *) transition. The nature of the electronic transitions that form the long-wavelength absorption band, regardless of their orbital nature, are localized on the benzene fragment of sulfaguanidine and are formed with the participation of the nitrogen atom N_7_ of the amino group (Figure 1a).

The centers of the molecule, which have the proton acceptor ability and determine the sites for the formation of H-bonds, were obtained from the calculation of the magnitude and coordinates of the minima of the electrostatic potential. The highest proton acceptor capacity in sulfaguanidine is in the oxygen atoms of the sulfonic group (U (O_9_) = −604 kJ/mol and U (O_10_) = −590 kJ/mol) (Figure 3b). It should be noted that there is a minimum of the electrostatic potential near the nitrogen atom (U (N_11_) = −450 kJ/mol) (Figure 1a). However, the approach for this center for a water molecule to form an H-bond is sterically difficult. In the case of a protonated complex, this situation persists.

The action of drugs in the body, the drugs’ metabolism and the hydrophilic and hydrophobic interactions between the cell membranes and the drugs at the first stage of interaction are determined by electrostatic interactions [33]. Thus, the pharmacological activity of medicinal compounds depends on the distribution and the change in the charges of these molecules. For this reason, an analysis of the results of calculating the effective charges of sulfaguanidine fragments and their changes in H-bonded complexes will be extremely useful. Table 4 shows the calculated effective charges on the active fragments of sulfaguanidine and its complexes.

### 2.2. Discussion

As noted above, the physiological activity of sulfaguanidine on a proton-donor medium [4] is determined by the properties of the aniline fragment. Our calculation showed that the addition of the sulfone and guanidine fragments to aniline in the S_0_ state practically does not change the charge of the amino group of the aniline fragment in sulfaguanidine. The formation of an H-bonded complex reduces the electron acceptor properties of the amino group of the aniline fragment and enhances the donor properties of the phenyl fragment, which, already being in the isolated molecule, makes the aniline fragment of sulfaguanidine an acceptor (Table 4). Thus, the addition of sulfone and guanidine fragments affects the properties of the biologically significant aniline fragment via the enhancement of the donor properties of the benzene fragment. The protonation of sulfaguanidine enhances this process (Table 2 and Table 4).

Upon excitation to the S_1_, S_2_, and S_3_ states, the amino group of the aniline fragment of sulfaguanidine becomes a particularly strong electron density donor. The formation of the complex and, especially, the protonation enhances the electron-donor properties of both the amino group and the aniline fragment of sulfaguanidine as a whole (Table 4). During the formation of H-bonds, the intrinsic charge of water molecules is weakly involved in the exchange of electron density between fragments of the sulfaguanidine molecule. Nevertheless, the formation of H-bonds contributes to an increase in the effective charge transferred between the fragments. It can be assumed that the initial interaction of an antibiotic in biological matter may be the binding of a positive aniline fragment of an antibiotic with negative sites on the surface of tubular cells.

### 2.3. Fluorescence Quantum Yield of Sulfaguanidine

Figure 4 shows the fluorescence spectra of sulfaguanidine and phenol in water at a concentration of 0.01 mM. According to Formula (2), the fluorescence quantum yield of sulfaguanidine is 0.16.

The Stokes shift of fluorescence for sulfaguanidine is large and amounts to 9700 cm^−1^, which indicates significant changes in the structure of the molecule in the fluorescent state. The experimental determination of the geometry of a molecule in the fluorescent state of polyatomic molecules is practically impossible. However, it can be determined by calculation, albeit partially.

It is known that there is a linear dependence between the length of a chemical bond and its electron density (population), from which it is possible to calculate the change in a chemical bond in an excited state if the change in its population during an electronic transition is known (according to Mulliken [36]). When calculating the lengths of chemical bonds in the backbone of molecules in the fluorescent state, the following formula was used:(1)$$\Delta R_{AB}^{*}=-k\Delta P_{AB}^{*}$$
where $\Delta P_{AB}^{*}$ is the change in the bond population upon excitation with respect to the ground state, and the coefficient *k* = 0.46 was obtained from the change in the C–C bond length during the S_0_→S_1_ transition in benzene [41].

Since the excitation of sulfaguanidine fluorescence occurs at the maximum of the absorption band, it should be expected that the fluorescent state will acquire the geometry of just this state. As shown above, the long-wavelength absorption band of sulfaguanidine is localized on the aniline fragment, and the largest changes in the geometry of the molecule occur in the S_3_ (ππ *) state, while changes in the chemical bond lengths of the backbone of the sulfanilamide fragment of this state are either insignificant or absent altogether. Therefore, Table 5 shows the data on the change in bond lengths only for the aniline fragment of sulfaguanidine. An analysis of the data in Table 5 showed that, in the electronic state S_3_ (ππ *), there are noticeable differences in the bond lengths of this fragment compared to the ground state of the molecule. In this case, changes in the bond lengths make the phenyl structure close to quinoid; i.e., in the S_3_ (ππ *) state, the phenyl aniline fragment of the molecule has a quasi-quinoid structure.

The calculation of the energy of the fluorescent state of sulfaguanidine with the quasi-quinoid structure of the phenyl fragment leads to such a change in the energies of electronic states that the state S_2_ (πσ *) becomes fluorescent (Table 6). In the case of such an orbital nature of the fluorescent state, it is problematic to observe radiation with a noticeable quantum yield, with rare exceptions. Our calculation of the fluorescence quantum yield yields a quantum yield less than 10^−4^. In this case, such a quantum yield takes place both for complexes and for cations, although the value of the Stokes shift is close to the experimental result (Table 7).

Experimentally, the value of the Stokes shift is the difference between the maxima of the absorption and fluorescence bands. In our opinion, in the case of the calculations, if the absorption band is formed by several electronic transitions, which takes place in the case of sulfaguanidine, the Stokes shift should be counted from the electronic transition that forms the maximum of the absorption band. For an isolated molecule, this is the 38,770 cm^−1^ electronic transition (Table 6).

A study [42] based on the analysis of the vibrational structure of the first absorption band of aniline vapor showed that the nitrogen atom of the amino group leaves the phenyl plane by about 0.98 Å [42]. The calculation of the energy of a fluorescent state with a quinoid structure and a pyramidal structure of the amino group (Table 7) shows a noticeable reduction in the energy of the fluorescent state, making the result close to the experiment’s not only in energy but also in the value of the radiative decay rate constant (Table 7).

The removal of the quasi-quinoid phenyl from the plane improves the agreement between the experimental and calculated fluorescence quantum yields and slightly changes the Stokes shift. The best agreement with the experiment is observed for the H-complexes of sulfaguanidine.

This is understandable since the quantum yield has been experimentally determined for an aqueous solution. In all structures, the main channel for the deactivation of absorbed energy is expected to be via the singlet–triplet conversion channel (Table 7).

### 2.4. Phosphorescence of Sulfaguanidine in Water

For a more detailed understanding of the relationship between the functional groups and fragments of the sulfaguanidine molecule and spectral manifestations, we experimentally and theoretically studied the spectra of photoluminescence and the excitation of fluorescence and phosphorescence of sulfaguanidine in water (Figure 5). It should be noted that sulfaguanidine has unique spectroscopic characteristics. In the presence of intense fluorescence and phosphorescence, it still has a large Stokes shift of about 10,000 cm^−1^, which means that the spectral properties of sulfaguanidine depend on the environment. The interaction of the antibiotic with the solvent will depend on the dielectric constant of the surrounding continuum.

Attention is drawn to the broadened and blurred nature of the phosphorescence excitation spectrum of sulfaguanidine in water with maxima in the region between 235 and 275 nm (Figure 6). According to [43], we assumed that at 77 K the H-bonds are weakened, which means that the electron conjugation chain is weakened and the fragmentation of the sulfaguanidine molecule is enhanced. The quantum chemical modeling of H-bond weakening was performed by varying the distance between the oxygen atoms of sulfaguanidine and H_2_O molecules. At each stage, the absorption spectra were calculated, and the contribution of configurations involving the molecular orbitals of the aniline fragment to the orbital nature of the excited states was calculated. An analysis of the orbital nature of electronic transitions to the S_1_ and S_3_ states, which make the largest contribution to the long-wavelength absorption band of sulfaguanidine, showed that S_0_→S_1_ and S_0_→S_3_ (or S_4_) transitions are formed to a greater extent by the molecular orbitals of the aniline fragment. For example, for the sulfaguanidine + 2H_2_O [44] complex in the S_1_ and S_3_ states, the contribution of configurations involving the aniline fragment increases: the decomposition coefficient for the main configuration of the aniline fragment of the S_1_ and S_3_ states varies from 0.622–640 (at a distance of -O…H_2_O = 1.74 Å) to 0.812–0.853 (at a distance of -O…H_2_O = 2.94 Å). This confirms our assumption that the role of the spectral manifestation of the aniline fragment increases with the weakening of the hydrogen bond system.

An analysis of the intercrossing conversion constants showed that the values of the matrix elements of the spin–orbital interaction for complexes of sulfaguanidine with water have maximum values for S_1_ − T_4_ = 2.88 cm^−1^ and S_3_ − T_4_ = 3.44 cm^−1^. Thus, the main channels for the transition of the excitation energy to triplet states and the formation of phosphorescence are S_1_~ > T_4_~ > T_1_→S_0_ and S_3_ (or S_4_) ~ > T_4_~ > T_1_→S_0_.

An analysis of the quantum chemical calculation allows us to conclude that the main contribution to the formation of phosphorescence is made by the aniline fragment of the sulfaguanidine molecule. This manifests itself in the localization of the electronically excited state on the aniline fragment.

A change in the spatial configuration of the molecular structure of the drug affects the position and intensity of the absorption, fluorescence and phosphorescence bands of sulfaguanidine. These effects are associated with the formation of a complex, a change in the role of various fragments of sulfaguanidine in the formation of absorption bands and the fluorescence and phosphorescence of the molecule. The shift of the absorption band maximum to shorter wavelengths can be caused by the large rotations of the aniline fragment relative to those of the sulfonyl and guanidine fragments. The rotation led to a fairly complete elimination of the π-π interaction of individual fragments, and the absorption spectrum became similar to the sum of the spectra of the constituent parts of the molecule. Although the interaction with the environment proceeds via the sulfonyl fragment, the contribution of the aniline fragment began to play a decisive role in the formation of spectral properties during complex formation or intense phosphorescence. The degree of conjugation between fragments for sulfaguanidine changes with a decrease in the coplanarity of the molecule. The appearance of a characteristic band at shorter wavelengths in the phosphorescence excitation spectrum becomes clearer as the degree of deviation from the coplanarity changes.

## 3. Materials and Methods

### 3.1. Experimental and Object

The object of our study is N-aminobenzenesulfonylguanidine monohydrate (sulfaguanidine) (Figure 1), which was synthesized by the commercial company Sigma-Aldrich (Si-Al article S8751-25G). White fine-crystalline powder of sulfaguanidine is slightly soluble in water. To obtain a matrix solution at a concentration of 1 mM, a dry sample was dissolved in distilled water using an ultrasonic stirrer.

Absorption spectra of sulfaguanidine in water were recorded on a VARIAN Cary 5000 Scan UV-VIS-NIR spectrophotometer (AgilentTech., USA-Netherlands-Australia) at room temperature in the range of 200 ÷ 500 nm. A quartz cuvette with an optical path length of 1 cm was used for measurements. The position of the electronic bands, which manifest themselves only as hidden maxima and fuzzy inflections in the absorption spectrum, was obtained by derivative spectrophotometry. This method is based on the same principles as conventional spectrophotometry; however, the analytical signal is not the optical density but its n-order derivative (usually with respect to the wavelength). Differentiation of the spectrum makes it possible to more clearly determine the position of the maximum of the absorption band. It also narrows the bands and makes it possible to determine substances that absorb at close wavelengths, the initial spectra of which partially overlap each other. According to this technique, it was possible to isolate electronic transitions in the experimental absorption spectra of sulfaguanidine in water. The absorption wavelength measurement error is 1 nm.

Fluorescence and phosphorescence spectra were recorded using standard methods on a VARIAN Cary Eclipse spectrofluorometer (AgilentTech., USA-Netherlands-Australia) at room temperature and at 77 K in the range of 200 ÷ 800 nm.

Calculations of the electronic absorption spectra of the compositions of the studied molecule, aniline and their complexes with water were performed by the semiempirical quantum chemical INDO method (e.g., neglect of diatomic differential overlap) using spectroscopic parameterization [45] and a software package developed in the Department of Molecular Photonics of the Siberian Institute of Physics and Technology at Tomsk State University. A significant advantage of this software package over modern programs for semiempirical methods of quantum chemistry is the ability to calculate the rate constants of intramolecular nonradiative processes of absorbed energy conversion (intersystem crossing and internal conversion) [46]. The software package makes it possible to calculate the molecular electrostatic potential U (kJ/mol) and the coordinates of the place of its maximum value [47,48], which makes it possible to reasonably simulate a complex with H-bonds.

### 3.2. Calculation of the Theoretical Quantum Yield Using Quantum Chemistry

The obtained wave functions were also used to calculate the rate constants of photophysical processes: internal and singlet–triplet conversions. The expression of the internal conversion rate constant was proposed by V.G. Plotnikov [44,49,50] and modernized by G.V. Mayer and V.Ya. Artyukhov [51,52]. The theoretical quantum yield was calculated by us from the following relation:(2)$$\varphi =\frac{k_{r}}{(k_{r}+k_{IC}+k_{ST)}},$$
where $k_{r}$, $k_{IC}$ and $k_{ST}\;$are rate constants (probabilities) of radiative decay processes, internal (between states of the same multiplicity) nonradiative conversion and intersystem crossing (or singlet–triplet, between states of different multiplicity) nonradiative conversion, respectively. The results obtained were compared with the experimental spectral-luminescent characteristics of sulfaguanidine.

### 3.3. Measurement of the Fluorescence Quantum Yield from a Standard

We determined the fluorescence quantum yield of sulfaguanidine experimentally by the relative method using Formula (2):(3)$$\varphi_{x}=\varphi_{et}\times \frac{S_{x}}{S_{et}}\times \frac{D_{et}}{D_{x}}\times \frac{n_{x}^{2}}{n_{et}^{2}},$$
where $S_{x}$ is area under the curve of the fluorescence spectrum of the test sample; *S_et_* is the area under the curve of the standard fluorescence spectrum; *D_et_* is the optical absorption density of the standard solution at the fluorescence excitation wavelength; *D_x_* is the optical absorption density of the test substance solution at the fluorescence excitation wavelength; *n_x_* is the refractive index of the solvent for the test substance; *n_et_* is the refractive index of the solvent for the standard; x and et, the sample and the standard are indicated.

The first step in measuring relative quantum yield is the choice of a reference. In this work, the standard was a solution of phenol in water, of which the fluorescence quantum yield is 0.032 [53]. Then, an aqueous solution of sulfaguanidine was prepared. The absorption spectra of phenol and sulfaguanidine in water were measured at a concentration of 0.01 mM. Finally, the fluorescence spectra of these solutions were measured to determine the area under them. For excitation, a wavelength of 250 nm was chosen since, when excitation occurs at different wavelengths, it is necessary to introduce corrections related to the unequal lamp intensity, which is an additional source of error.

## 4. Conclusions

On the basis of the experimental data and the quantum chemical calculations of the electronic structure of aniline and sulfaguanidine, changes in electronic absorption and fluorescence spectra and the effective charge distribution over fragments during the formation of H-bonded and protonated complexes are analyzed. It has been established that the oxygen atoms of the sulfonic fragment dominate in the complex formation and protonation of the sulfaguanidine molecule.

It has been shown that the sulfone and guanidine fragments presumably make the biologically active sulfaguanidine fragment via the benzene ring of the aniline fragment due to the donation of electron density. The formation of complexes with an H-bond or protonation enhances this property not only in the ground state but also in some electronically excited states.

Although deviations from standard accuracy levels for bioanalytical assays have long been accepted, narrow therapeutic ranges require further improvement. Deviation values can only be reduced by calibrating analytical instruments with standardized reference materials. Environmental effects and metabolic drug degradation are two known sources of emerging inaccuracies. In fact, these errors are one of the biggest problems in therapeutic drug monitoring because only a few standardized samples are available to manufacturers for assessing serum antibiotic residues, including LC-MS and immunoassays, which are considered routine. Spectral methods in clinical practice could take their rightful place in certified laboratories for the same antibiotic along with LC-MS and become one important part of the process of the therapeutic monitoring of drugs in biological objects and food products. In our opinion, the existing gap between clinical and scientific data is a problem for humanity, the solution of which lies in close cooperation between academic, industrial and medical partners.

## Figures and Tables

**Figure 1 molecules-28-04159-f001:**
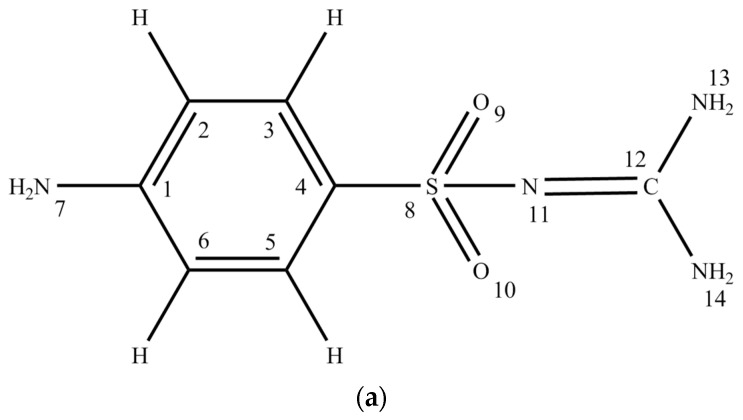
Structures of sulfaguanidine and its complexes: sulfaguanidine (**a**); sulfaguanidine + 2H_2_O (**b**); sulfaguanidine + 2H^+^H_2_O (**c**).

**Figure 2 molecules-28-04159-f002:**
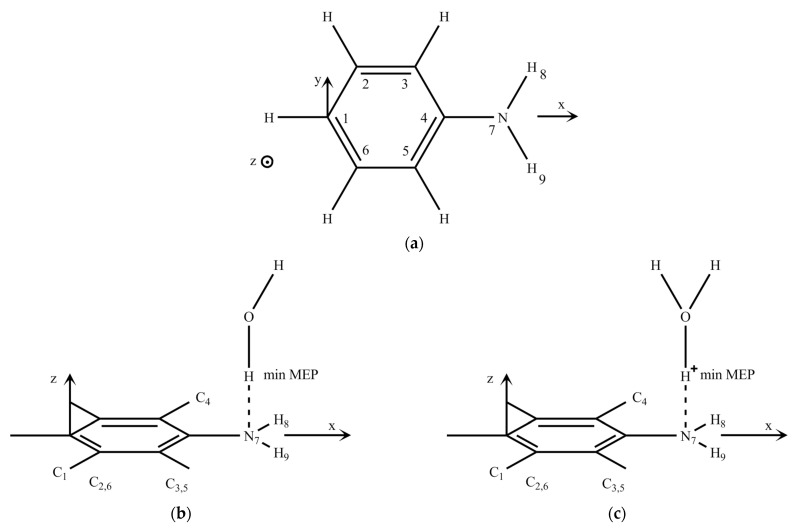
Structures of aniline and its complexes: aniline (**a**); aniline + H_2_O (**b**); aniline + H^+^H_2_O (**c**).

**Figure 3 molecules-28-04159-f003:**
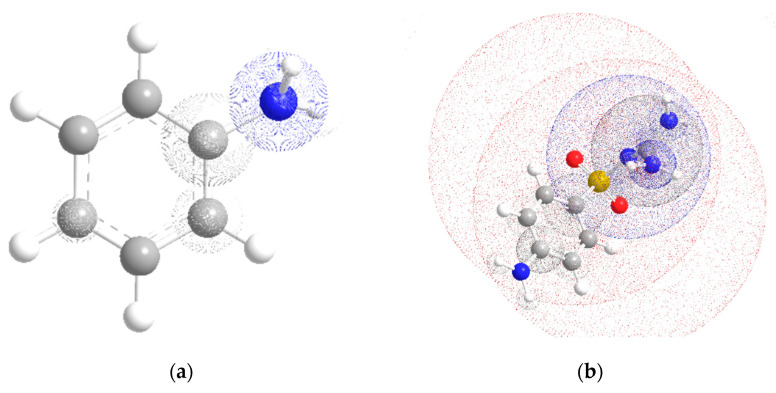
Calculated molecular electrostatic potential maps for aniline (**a**) and sulfaguanidine (**b**). Different colors indicate different electronegative regions.

**Figure 4 molecules-28-04159-f004:**
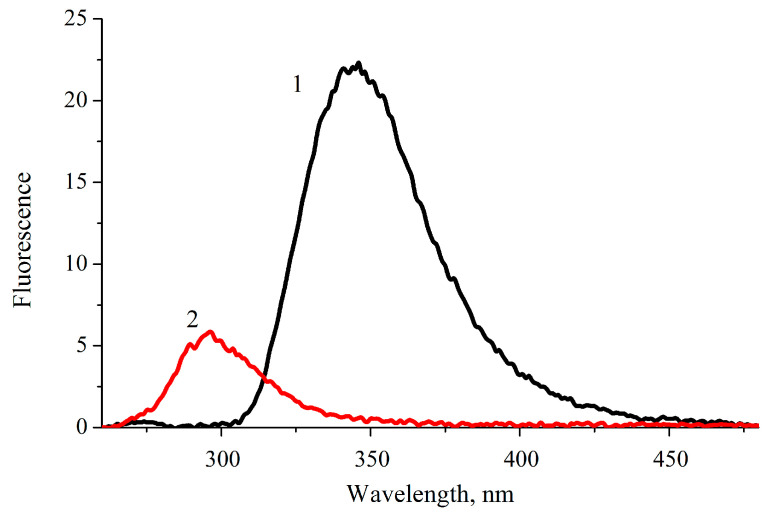
Fluorescence spectra of sulfaguanidine (1) and phenol (2) in water. The fluorescence excitation wavelength is 250 nm.

**Figure 5 molecules-28-04159-f005:**
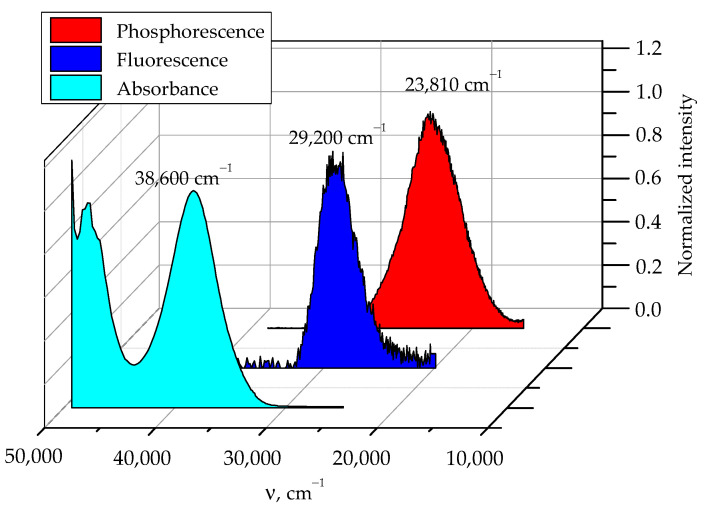
Normalized spectra of sulfaguanidine in water: absorption (cyan); fluorescence (blue); phosphorescence (red). The fluorescence excitation wavelength is 250 nm. The emission wavelength for fluorescence excitation is 350 nm.

**Figure 6 molecules-28-04159-f006:**
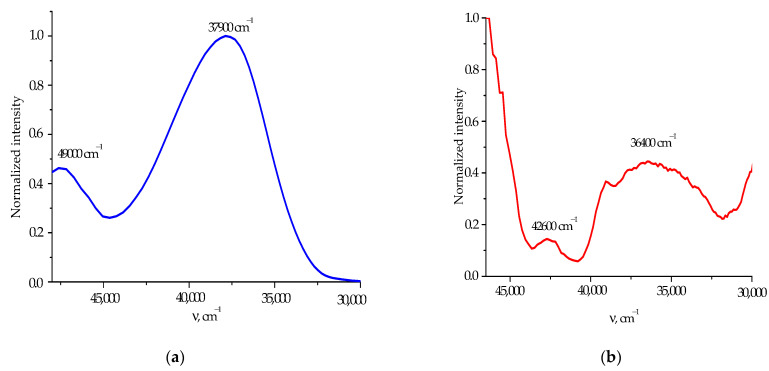
Normalized spectra of sulfaguanidine in water: fluorescence excitation (**a**); phosphorescence excitation (**b**).

**Table 1 molecules-28-04159-t001:** Spectral absorption characteristics of aniline, aniline in an H-bonded complex and aniline in a protonated form.

State	Structure of Aniline					
	Flat Structure			Nitrogen Atom Out of Plane, Z_N_ = 0.25 Å		
	Type	ν, cm^−1^ (λ, nm)	*f*	Type	ν, cm^−1^ (λ, nm)	*f*
Isolated molecule						
S_1_	ππ *	35,460 (282)	0.06	ππ *	36,460 (274)	0.01
S_2_	ππ *	39,550 (253)	0.06	ππ *	39,840 (250)	0.13
S_4_	-	-	-	ππ *	46,800 (214)	0.48
S_6_	ππ *	48,230 (207)	0.62	πσ *	48,990 (204)	0.14
S_7_	ππ *	48,470 (206)	0.74	ππ *	49,110 (203)	0.72
Aniline + H_2_O						
S_1_	ππ *	35,890 (279)	0.04	ππ * + πσ *	36,730 (272)	0.01
S_2_	ππ *	40,410 (248)	0.17	ππ *	40,780 (245)	0.14
S_4_	-	-	-	πσ *	47,200 (211)	0.44
S_6_	ππ *	48,410 (206)	0.62	ππ *	49,000 (204)	0.79
S_7_	ππ *	48,470 (206)	0.76	ππ * + πσ *	49,330 (203)	0.17
Aniline + H^+^H_2_O						
S_1_	ππ *	36,960 (270)	0.02	ππ *	37,100 (270)	0.00
S_2_	ππ *	42,410 (236)	0.14	ππ *	41,960 (238)	0.11
S_4_	-	-	-	ππ * + πσ *	46,940 (213)	0.31
S_5_	ππ *	48,880 (205)	0.74	ππ *	49,380 (202)	0.82
S_6_	ππ *	49,120 (204)	0.79	ππ * + πσ *	49,740 (201)	0.39

Note: *f*—one-electron transition oscillator strength.

**Table 2 molecules-28-04159-t002:** Distribution of effective charges (q_ef_, e) in aniline, aniline in an H-bonded complex and aniline in a protonated form.

Fragment	Flat Aniline			Aniline with Nitrogen Atom Out of Plane, Z_N_ = 0.25 Å		
	S_0_	S_1_	S_2_	S_0_	S_1_	S_2_
Isolated molecule						
–NH_2_	−0.045	0.143	0.135	−0.084	0.004	0.029
–C_6_H_5_	0.045	−0.143	−0.135	0.084	−0.004	−0.029
Aniline + H_2_O						
–NH_2_	−0.061	0.128	0.131	−0.082	0.009	0.116
–C_6_H_5_	0.091	−0.102	−0.105	0.115	−0.020	−0.012
H_2_O	−0.031	−0.026	−0.031	−0.033	−0.028	−0.026
Aniline + H^+^H_2_O						
	S_0_	S_1_	S_3_	S_0_	S_1_	S_3_
–NH_2_	−0.104	0.034	−0.275	−0.110	−0.094	−0.079
–C_6_H_5_	0.120	0.137	0.288	0.175	0.220	0.148
H^+^H_2_O	0.985	0.828	0.983	0.934	0.874	0.931

**Table 3 molecules-28-04159-t003:** Calculated and experimental absorption spectra of H-bonded and protonated sulfaguanidine complexes.

Calculation				Experiment	
State	*E_i_*, cm^−1^	λ, nm	*f*	*ν*_max,_ cm^−1^	λ_max_, nm
Sulfaguanidine + 2H_2_O					
S_1_ (ππ *)	35,240	284	0.05	34,500	290
S_2_ (πσ *)	38,450	260	0.00	38,600	259
S_3_ (ππ *)	39,110	256	0.40		
S_9_ (ππ * + σσ *)	47,330	211	1.00	48,100	208
S_10_ (ππ *)	47,830	209	0.44		
S_11_ (σσ *)	48,269	207	0.10		
Sulfaguanidine + 2H^+^H_2_O					
S_1_ (ππ *)	34,680	288	0.06	34,500	290
S_2_ (πσ *)	35,300	283	0.01		
S_3_ (ππ *)	38,300	261	0.47	39,100	256
S_8_ (ππ *)	47,230	212	0.51	48,350	207
S_9_ (ππ * + σσ *)	47,780	209	0.97		
S_10_ (ππ * + σσ *)	50,060	200	0.12	50,000	200
S_17_ (σσ *)	53,850	186	0.14		

Note: *E_i_* is the electronic transition energy, *ν* is the wave number, λ is the wavelength corresponding to the electronic transition, and *f* is the strength of the electronic transition oscillator.

**Table 4 molecules-28-04159-t004:** Changes in effective charges (q_ef_, e) on fragments of H-bonded and protonated sulfaguanidine complexes in various S_n_ electronic states.

Fragment, Group	Electronic State				
	S_0_	S_1_	S_3_	S_9_	S_10_
Sulfaguanidine					
Aniline:	0.081	0.048	0.075	0.019	0.048
NH_2_	−0.044	0.134	0.130	−0.027	−0.014
C_6_H_4_	0.125	−0.086	−0.055	0.046	0.062
Sulfonic	0.001	0.022	0.019	0.087	0.040
Guanidine:	−0.080	−0.070	−0.089	−0.103	−0.088
N_11_=C_12_	−0.074	−0.065	−0.085	−0.124	−0.085
2NH_2_	−0.006	−0.005	−0.004	0.021	−0.003
Sulfaguanidine + 2H_2_O					
Aniline:	0.092	0.068	0.076	−0.118	0.050
NH_2_	−0.040	0.134	0.149	−0.036	−0.004
C_6_H_4_	0.132	−0.066	−0.073	0.154	0.054
Sulfonic	0.063	0.079	0.083	0.153	0.142
Guanidine:	−0.070	−0.061	−0.075	0.046	−0.110
N_11_=C_12_	−0.078	−0.071	−0.086	−0.093	−0.121
2NH_2_	0.008	0.009	0.013	0.139	0.011
2H_2_O	−0.087	−0.086	−0.085	−0.084	−0.083
Sulfaguanidine + 2H^+^H_2_O					
	*S* _0_	*S* _1_	*S* _3_	*S* _8_	*S* _9_
Aniline:	0.170	0.205	0.339	0.349	0.096
NH_2_	−0.001	0.240	0.256	0.050	0.005
C_6_H_4_	0.171	−0.035	0.083	0.299	0.091
Sulfonic	−0.123	−0.121	−0.142	−0.136	−0.012
Guanidine:	−0.030	−0.066	−0.178	−0.166	0.064
N_11_=C_12_	−0.043	−0.073	−	−0.159	−0.112
2NH_2_	0.013	0.007	−0.005	−0.007	0.048
2H + H_2_O	1.983	1.983	1.983	1.983	1.981

**Table 5 molecules-28-04159-t005:** Chemical bond lengths (R, Å) of the backbone of the aniline fragment of sulfaguanidine in the ground state and in the S_3_ (ππ *) state.

*R_Si_*	Atom Number in Bonds							
	1–2	2–3	3–4	4–5	5–6	6–1	1–7	4–8
*R_S*0*_*	1.40	1.40	1.40	1.40	1.40	1.40	1.409	1.783
Isolated molecule								
*R_S*3*_*	1.466	1.390	1.489	1.493	1.393	1.470	1.402	1.806
Sulfaguanidine + 3H_2_O								
*R_S*3*_*	1.463	1.388	1.480	1.487	1.382	1.470	1.397	1.794
Sulfaguanidine + 2H^+^H_2_O								
*R_S*3*_*	1.460	1.390	1.464	1.497	1.397	1.465	1.397	1.803

Note: The numbering of atoms of chemical bonds corresponds to Figure 1a.

**Table 6 molecules-28-04159-t006:** The absorption and fluorescence energy dependence of the sulfaguanidine structure.

Absorbance			Fluorescence					
			The Quasi-Quinoid Structure with the Nitrogen Atom in the Phenyl Plane			The Quasi-Quinoid Structure with the Nitrogen Atom Outside the Phenyl Plane		
State	*E_i_*, cm^−1^	*f*	State	*E_i_*, cm^−1^	*f*	State	*E_i_*, cm^−1^	*f*
*S*_1_ (ππ *)	35,250	0.045	*S*_1_ (πσ *)	28,910	0.005	*S*_1_ (ππ *)	28,920	0.389
*S*_2_ (πσ *)	38,390	0.035	*S*_2_ (ππ *)	30,850	0.578	*S*_2_ (πσ *)	32,530	0.035
*S*_3_ (ππ *)	38,770	0.307	*S*_3_ (ππ *)	33,120	0.064	*S*_3_ (ππ *)	33,080	0.006

Note: *E_i_* is the state energy; *f* is the electronic transition oscillator strength.

**Table 7 molecules-28-04159-t007:** Fluorescence characteristics of sulfaguanidine, its complexes and its cationic forms.

Calculation						Experiment *		
*E_fl_*, cm^−1^ (nm)	Δ*ν*_ST_, cm^−1^	*k_r_*, s^−1^	*k_IC_*, s^−1^	*k_ST_*, s^−1^	φ	*E_fl_*, cm^−1^ (nm)	Δ*ν*_ST_, cm^−1^	φ
Isolated molecule						28,570 (350)	9800	0.16
28,920 (352)	10,350	2 × 10^8^	7 × 10^4^	1 × 10^10^	0.02			
Sulfaguanidine + 3H_2_O								
28,210 (354)	10,360	1 × 10^8^	2 × 10^5^	6 × 10^8^	0.18			
Sulfaguanidine + H_2_O + 2(H^+^H_2_O) *q* = +2*e*								
28,360 (353)	9510	2 × 10^8^	1 × 10^5^	5 × 10^9^	0.03			

Note: * Data given for aqueous solution. *E_fl_* is the energy of the fluorescent state; Δ*ν*_ST_ is the Stokes shift.

## Data Availability

The materials and datasets generated and analyzed during the current study are publicly available from the corresponding author on reasonable request.

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
