# Peer review of "Nature of Luminescence and Pharmacological Activity of Sulfaguanidine"

_molecules, 2023, doi:10.3390/molecules28104159_

Round 1

Reviewer 1 Report

The authors presented a manuscript in which they evaluated the fluorescence of sulfaguanidine in aqueous solutions. Sulfaguanidine - an organic chemical compound, an amide of sulfanilic acid and guanidines. Used as a drug from the group of sulfonamides. Currently, very rarely used in bacterial infections of the gastrointestinal tract, gastritis, enteritis. It has a bacteriostatic effect on the bacteria of the digestive tract. currently, sulfonamides are used primarily in connection with the huge and growing resistance to antibiotics. For methodological reasons, the work is presented correctly, but the aim of the article is to present the idea to a wide audience. Work may be of interest not only to pharmacists or scientists dealing with molecular pharmacy, but it may be interesting for practicing physicians, especially clinical pharmacologists.
Therefore, it would be advisable:
- extension of the introduction, adding a paragraph emphasizing the practical use of the presented aspects, e.g. in drug concentration-monitored therapy (and in relation to classic drugs, therapeutic drug monitoring of digoxin or carbamazepine were published interesting 20 years observational studies). In the paragraph, they should refer to the possibility of using e.g. TDM by comparing it to "classic" drugs
- the discussion should include a short paragraph describing the practical aspect: when, how to think about this type of activities, whether in clinical and veterinary medicine or maybe only in forensic medicine. Please look through the prism of current indications for use. This type of paragraph will make the manuscript more interesting for readers

Author Response

Point-by-point response to Reviewer 1:

  1. …extension of the introduction, adding a paragraph emphasizing the practical use of the presented aspects, e.g. in drug concentration-monitored therapy (and in relation to classic drugs, therapeutic drug monitoring of digoxin or carbamazepine were published interesting 20 years observational studies). In the paragraph, they should refer to the possibility of using e.g. TDM by comparing it to "classic" drugs…

In Introduction the next was added:

Modeling and prediction of the properties and biological activity of molecules depends on the choice of molecular representation [110]. Decades of research have been required to establish structure-activity and structure-property relationships for organic compounds. Understanding these relationships opens the way to improving the predictive modeling of biological activity and the properties of small molecules for the synthesis of new drugs. The monitoring of therapeutics in biological systems and foodstuffs holds great potential for improving patient outcomes and dramatically reducing healthcare costs [1101-1103]. Although it has been on the radar of the scientific community for nearly two decades, sensor-based approaches have not yet been widely adopted into clinical laboratory diagnostics, perhaps due to a gap between the scientific, industrial, and medical communities. Looking ahead, molecular understanding of the spectral properties of molecules will become increasingly important and will provide a solid basis for future predictive tasks associated with the determination of drugs in serum and foods.

[110] K.V. Chuang, L.M. Gunsalus, and M.J. Keiser/ Learning Molecular Representations for Medicinal Chemistry Miniperspective //J. Med. Chem. 2020, Vol. 63, 8705−8722. Doi: 10.1021/acs.jmedchem.0c00385

[1101] H.M. Dardeer, A.Toghan, M.E. A. Zaki, R.B. Elamary /Design, Synthesis and Evaluation of Novel Antimicrobial Polymers Based on the Inclusion of Polyethylene Glycol/TiO2 Nanocomposites in Cyclodextrin as Drug Carriers for Sulfaguanidine // Polymers. 2022. Vol.14. P. 227(20). doi: 10.3390/ polym14020227.

[1102] N.El Alami El Hassani, E. Llobet, L.-M. Popescu, M. Ghita, B. Bouchikhi b, N. El Bari /Development of a highly sensitive and selective molecularly imprinted electrochemical sensor for sulfaguanidine detection in honey samples //Journal of Electroanalytical Chemistry. 2018, Vol. 823. P.  647-655. doi:10.1016/j.jelechem.2018.07.011.

[1103] I.J.D. Priscillal, A.A. Alothman, S.-F. Wang, R. Arumugam /Lanthanide type of cerium sulfide embedded carbon nitride composite modified electrode for potential electrochemical detection of sulfaguanidine // Microchimica Acta. 2021. Vol. 188. P. 313(12). doi:10.1007/s00604-021-04975-y.

  1. …the discussion should include a short paragraph describing the practical aspect: when, how to think about this type of activities, whether in clinical and veterinary medicine or maybe only in forensic medicine. Please look through the prism of current indications for use. This type of paragraph will make the manuscript more interesting for readers…

In Discussion paragraph we added:

A change in the spatial configuration of the molecular structure of the drug affects the position and intensity of the absorption, fluorescence and phosphorescence bands of sulfaguanidine. These effects are associated with the formation of a complex, a change in the role of various fragments of sulfaguanidine in the formation of absorption bands, fluorescence and phosphorescence of the molecule. The shift of the absorption band maximum to shorter wavelengths can be caused by large rotations of the aniline fragment relative to the sulfonyl and guanidine fragments. The rotation led to a fairly complete elimination of the π-π interaction of individual fragments, and the absorption spectrum became similar to the sum of the spectra of the constituent parts of the molecule. Although the interaction with the environment proceeds through the sulfonyl fragment, the contribution of the aniline fragment began to play a decisive role in the formation of spectral properties during complex formation or intense phosphorescence. The degree of conjugation between fragments for sulfaguanidine changes with a decrease in the coplanarity of the molecule. The appearance of a characteristic band at shorter wavelengths in the phosphorescence excitation spectrum becomes clearer as the degree of deviation from coplanarity changes.

In Conclusion paragraph we added:

Although deviations from standard accuracy levels for bioanalytical assays have long been accepted, narrow therapeutic ranges require further improvement. Deviation values can only be reduced by calibrating analytical instruments with standardized reference materials. Environmental effects and metabolic drug degradation are two known sources of emerging inaccuracies. In fact, these errors are one of the biggest problems in therapeutic drug monitoring. Because only a few standardized samples are available to manufacturers for assessing serum antibiotic residues, including LC-MS and immunoassays, which are considered routine. Spectral methods in clinical practice could take their rightful place in certified laboratories for the same antibiotic along with LC-MS. And become one important part of the process of therapeutic monitoring of drugs in biological objects and food products. In our opinion, the existing gap between clinical and scientific data is a problem of humanity, the solution of which lies in close cooperation between academic, industrial and medical partners.

Thanks

Best regards

Prof. Olga Tchaikovskaya

May 15, 2023

Reviewer 2 Report

In this research, the author studied the fluorescent state of sulfaguanidine by using computational chemistry, electronic absorption, and luminescence methods. It was found that the sulfone and guanidine fragments may form biologically active sulfaguanidine fragments through the benzene ring of the aniline fragment. The formation of complexes with an H-bond or protonation enhances this property both in the ground state and in some electronically excited states.

There are still some errors in this manuscript which need to be carefully corrected before publication. If the following issues are well-addressed, this reviewer believes that the essential contribution of this paper is important for the usage of faguanidine as a drug compound.

  1. The first paragraph of the article is an introduction from a large research scope to a specific scientific frontier issue, which requires sufficient literature support. But this manuscript does not cite any papers, so the author needs to rewrite the first paragraph or recite the relevant literature.
  2. In line 99, is “in the range of 200÷500 nm” should be “in the range of 200 to 500 nm“? And also in line 113, should be “200 to 800 nm“?
  3. If an abbreviation appears for the first time in the text, the author should give the full name, as in line 115, “INDO method“.
  4. In line 249, it should be “Figure 1a“.
  5. In line 369, can the author explain why choose 77 K?
  6. The authors need to give more description or explanation to Figure 5
  7. The significance of this paper is not expounded sufficiently. The author needs to highlight this paper's innovative contributions in the conclusion section.
  8. Authors need to unify the format of all references, such as the volume, like Refs 12, 43; and whether to add DOI, like Refs 1,12, 13, 26.

Author Response

Point-by-point response to Reviewer 2:

  1. The first paragraph of the article is an introduction from a large research scope to a specific scientific frontier issue, which requires sufficient literature support. But this manuscript does not cite any papers, so the author needs to rewrite the first paragraph or recite the relevant literature.

In Introduction the references were added:

Modern computer technologies, bioinformatics and new experimental methods in the field of medicinal chemistry have provided acceleration and optimization of the process of finding new biologically active compounds [110]. Currently, the pharmaceutical industry widely uses molecular modeling methods to study the relationship between the structure and the activity of molecules. Existing quantum chemical methods helps to create new drugs by identifying the substituents and their location in the molecule necessary to obtain a pharmaceutical effect. Among amine derivatives, there are a number of well-known highly effective drugs [111]. Very often, amines are not used in a free form, but in the form of sulfonyl derivatives - sulfanilamide preparations used to treat infectious diseases, mainly of bacterial origin. The identification of the most probable conformations for binding the molecule to the environment includes two stages: the study of the entire conformational space of the molecular structure and the exact determination of the site of interaction. To search for the corresponding conformations of the molecule, systematic and stochastic search methods are used. These methods give variations in structural parameters (angles, bond lengths), gradually revealing the appropriate conformation [112].

[110] K.V. Chuang, L.M. Gunsalus, and M.J. Keiser/ Learning Molecular Representations for Medicinal Chemistry Miniperspective //J. Med. Chem. 2020, Vol. 63, 8705−8722. Doi: 10.1021/acs.jmedchem.0c00385

[111] M.R. Aouad, D.J.O. Khan, M.A. Said, N.S. Al-Kaff, N. Rezki, A.A. Ali, N. Bouqellah,  M. Hagar/Novel 1,2,3-Triazole Derivatives as Potential Inhibitors against Covid-19 Main Protease: Synthesis, Characterization, Molecular Docking and DFT Studies// ChemistrySelect. 2021. Vol. 6, Is. 14. P. 3468-3486. doi:10.1002/slct.202100522

[112] T.D.N. Luong, S.Nagpal, M. Sadqi , V. Muñoz/ A modular approach to map out the conformational landscapes of unbound intrinsically disordered proteins// Biophysics and computational biology. 2022. Vol. 119, Is. 23. P. e2113572119. doi.org/10.1073/pnas.2113572119

  1. In line 99, is “in the range of 200÷500 nm” should be “in the range of 200 to 500 nm“? And also in line 113, should be “200 to 800 nm“? – correct. “in the range of 200 to 500 nm“ – for absorbance spectra and “200 to 800 nm“ – for phosphorescence spectra
  2. If an abbreviation appears for the first time in the text, the author should give the full name, as in line 115, “INDO method“ – has been corrected  ‘the semiempirical quantum-chemical  INDO method (e.g., neglect of diatomic differential overlap)’
  3. In line 249, it should be “Figure 1a“ – has been corrected 
  4. In line 369, can the author explain why choose 77 K? – reference was added:

A.E. Khudozhitkov,  P. Stange, B. Golub,  D. Paschek, A. G. Stepanov, D. I. Kolokolov, R. Ludwig/Characterization of Doubly Ionic Hydrogen Bonds in Protic Ionic Liquids by NMR Deuteron Quadrupole Coupling Constants: Differences to H-bonds in Amides, Peptides, and Proteins// Angewandte Chemie International Edition. 2017. Vol. 56, Is. 45. P. 14310-14314. doi:10.1002/anie.201708340

  1. The authors need to give more description or explanation to Figure 5 – has been added: It should be noted that sulfaguanidine has unique spectroscopic characteristics. In the presence of intense fluorescence and phosphorescence, it still has a large Stokes shift of about 10000 cm-1, which means that the spectral properties of sulfaguanidine depend on the environment. The interaction of the antibiotic with the solvent will depend on the dielectric constant of the surrounding continuum.
  2. The significance of this paper is not expounded sufficiently. The author needs to highlight this paper's innovative contributions in the conclusion section. – in conclusion has been added

Although deviations from standard accuracy levels for bioanalytical assays have long been accepted, narrow therapeutic ranges require further improvement. Deviation values can only be reduced by calibrating analytical instruments with standardized reference materials. Environmental effects and metabolic drug degradation are two known sources of emerging inaccuracies. In fact, these errors are one of the biggest problems in therapeutic drug monitoring. Because only a few standardized samples are available to manufacturers for assessing serum antibiotic residues, including LC-MS and immunoassays, which are considered routine. Spectral methods in clinical practice could take their rightful place in certified laboratories for the same antibiotic along with LC-MS. And become one important part of the process of therapeutic monitoring of drugs in biological objects and food products. In our opinion, the existing gap between clinical and scientific data is a problem of humanity, the solution of which lies in close cooperation between academic, industrial and medical partners.

  1. Authors need to unify the format of all references, such as the volume, like Refs 12, 43; and whether to add DOI, like Refs 1,12, 13, 26. – we tried to make it possible to do so where possible

Thanks

Best regards

Prof. Olga Tchaikovskaya

May 15, 2023

Round 2

Reviewer 1 Report

the manuscript after the correction may be considered for publication in my opinion